# BRAF Modulates Lipid Use and Accumulation

**DOI:** 10.3390/cancers14092110

**Published:** 2022-04-23

**Authors:** Jacqueline A. Turner, Emily L. Paton, Robert Van Gulick, Davide Stefanoni, Francesca Cendali, Julie Reisz, Richard P. Tobin, Martin McCarter, Angelo D’Alessandro, Raul M. Torres, William A. Robinson, Kasey L. Couts, Isabel R. Schlaepfer

**Affiliations:** 1Division of Medical Oncology, Anschutz Medical Campus, University of Colorado School of Medicine, Aurora, CO 80045, USA; emily.paton@cuanschutz.edu (E.L.P.); robert.vangulick@colorado.edu (R.V.G.); william.robinson@cuanschutz.edu (W.A.R.); kasey.couts@cuanschutz.edu (K.L.C.); isabel.schlaepfer@colorado.edu (I.R.S.); 2Medical Scientist Training Program, Anschutz Medical Campus, University of Colorado School of Medicine, Aurora, CO 80045, USA; 3Department of Immunology and Microbiology, Anschutz Medical Campus, University of Colorado School of Medicine, Aurora, CO 80045, USA; raul.torres@cuanschutz.edu; 4Department of Biochemistry and Molecular Genetics, Anschutz Medical Campus, University of Colorado School of Medicine, Aurora, CO 80045, USA; davide.stefanoni@cuanschutz.edu (D.S.); francesca.cendali@cuanschutz.edu (F.C.); julie.haines@cuanschutz.edu (J.R.); angelo.dalessandro@cuanschutz.edu (A.D.); 5Divion of Surgical Oncology, Department of Surgery, Anschutz Medical Campus, University of Colorado School of Medicine, Aurora, CO 80045, USA; richard.tobin@cuanschutz.edu (R.P.T.); martin.mccarter@cuanschutz.edu (M.M.)

**Keywords:** BRAF, polyunsaturated, lipids, melanoma

## Abstract

**Simple Summary:**

BRAF is a serine/threonine kinase that is commonly mutated across cancers. The BRAF V600E mutation is targetable with kinase inhibitors; however, many patients eventually develop resistance. Recent evidence suggests that tumors harboring *BRAF* mutations may oxidize fatty acids for energy rather than utilizing aerobic glycolysis (the Warburg effect). Understanding the metabolism of cells harboring *BRAF* mutations may uncover targets to improve therapy response. We studied the effects of *BRAF* mutation and expression on metabolism. We found that cell expressing BRAF V600E were enriched with immunomodulatory lipids and have a metabolism that is distinct from cells expressing wild type BRAF. We also found that patients with melanoma who did not respond to BRAF-targeted therapy had plasma lipid profiles that were different from patients who responded to this therapy. Overall, our findings indicate that targeting lipid metabolism may be a potential alternative strategy to improve patient responses to BRAF-targeted therapies.

**Abstract:**

There is increasing evidence that oxidative metabolism and fatty acids play an important role in BRAF-driven tumorigenesis, yet the effect of *BRAF* mutation and expression on metabolism is poorly understood. We examined how *BRAF* mutation and expression modulates metabolite abundance. Using the non-transformed NIH3T3 cell line, we generated cells that stably overexpressed BRAF V600E or BRAF WT. We found that cells expressing BRAF V600E were enriched with immunomodulatory lipids. Further, we found a unique transcriptional signature that was exclusive to BRAF V600E expression. We also report that *BRAF V600E* mutation promoted accumulation of long chain polyunsaturated fatty acids (PUFAs) and rewired metabolic flux for non-Warburg behavior. This cancer promoting mutation further induced the formation of tunneling nanotube (TNT)-like protrusions in NIH3T3 cells that preferentially accumulated lipid droplets. In the plasma of melanoma patients harboring the *BRAF V600E* mutation, levels of lysophosphatidic acid, sphingomyelin, and long chain fatty acids were significantly increased in the cohort of patients that did not respond to BRAF inhibitor therapy. Our findings show BRAF V600 status plays an important role in regulating immunomodulatory lipid profiles and lipid trafficking, which may inform future therapy across cancers.

## 1. Introduction

The mitogen-activated protein kinase (MAPK) pathway is a kinase cascade with RAS-RAF-MEK-ERK kinase components and is essential for cellular growth and development. Approximately 30% of all human cancers have mutations in the MAPK pathway and approximately 7% are mutations in the RAF kinase gene, *BRAF* [1,2]. BRAF is a serine/threonine kinase that may be phosphorylated, dimerize, and signal downstream [3,4,5]. The most common mutation in *BRAF* is a GTG→GAG mutation at codon 600 that results in a valine to glutamic acid transition (*BRAF V600E*) [6]. The glutamic acid associates with a positively charged lysine residue in the N-lobe of the kinase to promote kinase closure and activity [7]. *BRAF* mutation also modulates the aspartate-phenylalanine-glycine (DFG) regulatory motif to adopt an active DFG-in conformation resulting in constitutive kinase activity [8]. BRAF V600E protomers can signal as monomers, dimers, or kinase-dead protomers and are not regulated by ERK feedback phosphorylation [9,10,11]. The *BRAF V600E* mutation is found across cancer types including >97% of hairy cell leukemias, 40–67% of melanomas, 36–69% of papillary thyroid tumors, 15–20% of low-grade pediatric tumors, and 5–17% of colorectal cancers [12,13,14,15,16]. Further, *BRAF V600E* oncogenic mutations in monocytes/macrophages etiologically contribute to the development of human inflammatory neoplasms such as in Erdheim–Chester disease [17].

Conventional therapies targeting the BRAF V600E mutation often have short-lived benefit since many tumors quickly acquire resistance [18]. Recent efforts have identified specific changes in metabolism that are associated with BRAF activity. SREBP-1 lipogenesis, PGC1α oxidative metabolism, MITF, and other transcriptional regulators play important roles in regulating BRAF activity and consequently sensitivity to BRAF inhibition [19,20,21,22,23]. These studies identify potential therapeutic strategies to improve sensitivity to BRAF inhibition and overcome resistance in BRAF mutated cancers. While these preliminary studies have promising results, we have yet to understand the precise mechanisms by which the *BRAF V600* mutation affects metabolism. Importantly, this information is essential to effectively modulate tumor metabolism and improve therapies for BRAF mutated cancers.

In this study, we introduced a *BRAF V600E* mutation into non-transformed fibroblast cells and found BRAF V600 status modulates metabolite abundancy. Specifically, the *BRAF V600E* mutation rewires metabolic flux for non-Warburg-like behavior and promotes accumulation of lipids. Immunomodulatory long chain polyunsaturated fatty acids (PUFAs) were enriched in cells expressing BRAF V600E. These metabolic changes were accompanied by transcriptional upregulation of *FAS*, *CPT1a*, *PPARγ*, and *SCLA27a1*. Further, *BRAF V600E* mutation changes the cellular phenotype and induces formation of long F-actin containing protrusions that preferentially accumulate lipid droplets. In patients with melanomas harboring the *BRAF V600E* mutation, immunomodulatory lipids and long chain fatty acids were significantly increased post-BRAF inhibitor (BRAFi) therapy in the non-responder cohort. Specifically, palmitic acid, adrenic acid, lysophosphatidic acid, and sphingomyelin were significantly increased post-treatment in the plasma from non-responders. Together, these data show BRAF V600 status plays an important role in determining the immunomodulatory lipid profile and lipid trafficking, which may inform future combination therapies to improve patient response to BRAF inhibitor therapy across cancers.

## 2. Materials and Methods

### 2.1. Plasmids and Gene Constructs

cDNA sequences for wild type *BRAF* were synthesized (GenScript Biotech, Piscataway, NJ, USA) and cloned in to the pLVX-EF1α-IRES-ZsGreen vector (Clonetech Laboratories Inc., Fremont, CA, USA) as previously used [24]. The *BRAF V600E* clone was generated from the wild type *BRAF* sequence using site directed mutagenesis (GenScript). Vectors were introduced and grown in One Shot^®^ Mach1^TM^ T1 (Clontech Laboratories Inc., San Jose, CA, USA) phage-resistant chemically competent *E. coli* (Invitrogen, Carlsbad, CA, USA). DNA was extracted using a HiSpeed^®^ Plasmid Midi Kit (Qiagen, Hilden, Germany).

### 2.2. Stable Gene Expression, Cell Lines, and Culture Conditions

Viral particles were produced by transfecting human embryonic kidney (HEK) 293T cells (ATCC, Manassas, VA, USA) with empty vector DNA and vectors with either wild type *BRAF* or *BRAF V600E* inserted (Clonetech Laboratories Inc.) using Lenti-X^TM^ Packaging Single Shots (VSV-G) (Clontech Laboratories Inc.). Virus was collected and filtered at 48 and 72 h. Mycoplasma tested and STR profiled NIH3T3 cells (ATCC) were infected with viral particles and 8 µg/mL of polybrene (Santa Cruz Biotechnology Inc., Dallas, TX, USA). Infected NIH3T3 cells were then sorted using fluorescence-activated cell sorting (FACS) by a MoFlo XDP cell sorter (Beckman Coulter, Brea, CA, USA). Cells were then mass cultured in RPMI growth media with 10% fetal bovine serum and 1% penicillin-streptomycin.

### 2.3. Mass Spectrometry, Metabolomics, and Lipidomics

Samples were collected from in vitro cultures or patient plasma samples. Cells were plated in 15 cm plates in triplicate with 300,000 cells per plate. After 72 h, cells and supernatant were harvested. One million cells per sample were pelleted. All samples were stored at −80 °C before being analyzed by liquid chromatography/tandem mass spectrometry as previously described [25]. In brief, metabolites from frozen cell pellets were extracted at 4 °C in the presence of 5:3:2 MeOH:MeCN:water (*v*/*v*/*v*) and the resulting supernatant was analyzed by a Thermo Vanquish UHPLC coupled to a Thermo Q Exactive mass spectrometer as previously described in detail [26]. Lipids from frozen cell pellets were extracted at 4 °C in the presence of methanol and the resulting supernatant was analyzed by a Thermo Vanquish UHPLC coupled to a Thermo Q Exactive mass spectrometer as previously described in detail [27]. All mass spectrometry data is provided in Appendix A.

### 2.4. Metabolic Flux Assay

Cells were plated at 20,000 cells per well of a 96-well plate in complete media and cultured in a CO_2_ incubator overnight. The sensor cartridge was placed in a utility plate that was loaded with 200 µL of XF Calibrant and hydrated in a non-CO_2_ incubator overnight. Oligomycin (75351, Sigma-Aldrich, St. Louis, MO, USA), FCCP ((4-(trifluoromethoxy) phenyl) carbonohydrazonoyl dicyanide, C2920, Sigma-Aldrich), and antimycin A (A8674, Sigma-Aldrich) + rotenone (R8875, Sigma-Aldrich) were prepared for a titration curve at concentrations of 5 µM, 10 µM, 100 µM, and 1000 µM in 25 mM glucose at pH 7.4. Drugs were loaded into the cartridge and the cartridge was run on a Seahorse XFe96 Analyzer with 96-well plates (Agilent Technologies, Santa Clara, CA, USA). A 10 µM drug concentration was determined as optimal for data collection with NIH3T3 cells (Appendix A). Supplemented fatty acids were purchased from Sigma-Aldrich, resuspended in ethanol for a stock solution of 10 mM, and stored at −80 °C. Drugs were prepared in either 25 mM glucose or 25 µM palmitic acid, oleic acid, or α-linolenic acid at pH 7.4 and cells were analyzed for OCR and ECAR. Analysis was performed using Seahorse Wave software (Agilent Technologies).

### 2.5. Immunofluorescence and Microscopy

Cells were cultured on a 22 mm glass coverslip placed in a 6-well plate with 20,000 cells per well. After 24 h, the cells were washed with PBS and fixed in 10% formalin for 20 min. The coverslips were stored in PBS at 4 °C until they were permeabilized with BD perm/wash buffer (554723, BD Biosciences, San Jose, CA, USA). Lipid droplets were stained using Nile red (1:500, 19123, Sigma-Aldrich) for 20 min at room temperature. F-actin was stained using the phalloidin-Alexa 488 antibody (1:1000, Thermo Fisher Scientific, Waltham, MA, USA) for 1 h at room temperature. Nuclei were stained using DAPI (1:1000, 62248, Thermo Fisher Scientific) for 5 min at room temperature. Coverslips were washed with PBS and mounted onto slides using mounting media (ab104135, Abcam, Cambridge, UK). Images were taken with an Olympus 1X83 V-TB190 inverted microscope at 400× using cellSens software and quantified using FIJI software (available online: https://imagej.nih.gov/ij/ and accessed on 7 July 2019). Total fluorescence was calculated from the red channel and normalized for the number of cells. Regions of interest (ROIs) were used to quantify regional cell fluorescence with a minimum of 15 measurements per condition.

### 2.6. Melanoma Patient Samples

Samples were collected from ten melanoma patients from the University of Colorado Hospital from 2008 to 2020. Plasma samples from melanoma patients were collected in green-top heparin tubes with pre- and post-treatment samples. These samples were collected directly from the hospital and processed immediately and frozen in a −80 °C freezer until use. These plasma samples were never thawed or moved in the interim. These samples were obtained as part of the International Melanoma Biorepository and Research Laboratory at the University of Colorado Cancer Center. Patients were consented under the approval from the Colorado Institutional Review Board (IRB# 05-0309). These patient studies were conducted according to the Declaration of Helsinki, Belmont Report, and U.S. Common Rule.

### 2.7. Sanger Sequencing and Quantitative Real-Time PCR

Genomic DNA was isolated using the DNeasy Blood and Tissue kit (Qiagen, Hilden, Germany). PCR was performed using GoTaq (Promega, Madison, WI, USA) with the following primer sequences specific for *BRAF*: 5′ CTCCAGCTTGTATCACCATCTC 3′; 5′ CTGGTCCCTGTTGTTGATGT 3′. Additional PCR primers are listed in Appendix A. PCR products were purified using the QIAquick PCR purification kit (Qiagen) and submitted to the Barbra Davis Center for sequencing using the BigDye Terminator Cycle Sequencing Ready Reaction kit version 3.1 (Applied Biosystems, Foster City, CA, USA). RNA was extracted from pelleted cells using the RNeasy Plus Mini Kit (Qiagen). Cells were homogenized by pipetting and on-column DNase I digest was performed using an RNase-free DNase I set (Qiagen). RNA was reverse transcribed into cDNA using the Verso cDNA Synthesis Kit (Thermo Fisher Scientific). Quantitative real-time PCR was carried out in triplicate using PowerUp SYBER Green master mix (Thermo Fischer Scientific) and was analyzed on the StepOne Plus real-time PCR system (Applied Biosystems). Primer sequences are listed in Appendix A (Sigma-Aldrich).

### 2.8. Western Immunoblotting

Cell lysis was carried out in cold RIPA buffer with added protease and phosphatase inhibitors (Thermo Fisher Scientific) for 10 min. Lysates were centrifuged at 13,000 rpm for 10 min. Fifty micrograms of protein was loaded into each well and separated by SDS-PAGE. Samples were transferred to nitrocellulose membranes. The following primary antibodies from Cell Signaling (Danver, MA, USA) and SC Biotechnology (Dallas, TX, USA) were used: phospho-BRAF (Ser 445, #2696), total BRAF (sc-166), and β-actin (#4970). LiCor (Lincoln, NE, USA) fluorescent anti-rabbit and -mouse secondary antibodies were used, and blots were imaged using a Li-Cor Odyssey.

### 2.9. Statistical Analyses

Experiments were performed in biological replicates. Results are expressed as the mean ± standard error of the mean. Direct comparisons were made using non-parametric analyses, ANOVA, and Student’s *t* Test. Cohort sizes were determined based on statistical and power considerations.

## 3. Results

### 3.1. BRAF Expression and Mutation Modulates Metabolic Profiles

To study how *BRAF* mutation affects metabolism, we generated stable cell lines overexpressing wild type BRAF (BRAF WT) or BRAF V600E using the NIH3T3 mouse fibroblast cell line (Appendix A). With this model, we assessed how BRAF V600 status and kinase activity affected basal metabolism using high resolution mass spectrometry to measure global metabolites. First, we performed an unsupervised principal component analysis (PCA) (Figure 1A). In this three-dimensional PCA, we found over 90% of variation in the data was observed for three principal components. Each of the triplicates clustered together, and cells overexpressing BRAF WT or BRAF V600E clustered separately from the parental or control cell lines. A hierarchical clustering analysis of 156 metabolites further showed BRAF WT and BRAF V600E cells clustered separately from the controls (Figure 1B). Interestingly, cells overexpressing BRAF WT had a metabolic profile that was more different from the controls than cells overexpressing BRAF V600E. Altogether, these data demonstrate BRAF V600E cells have a metabolic profile different from BRAF WT cells. To determine specific metabolic differences between BRAF WT and BRAF V600E cells, we performed a differential metabolite analysis to identify the top 50 metabolites most abundant in BRAF V600E cells (Figure 1C; enriched metabolites). We found lipids comprised the largest percentage of the top metabolites (% of top 50) and remained the most enriched metabolite class when normalized to the total number of metabolites measured (normalized %). In a separate analysis, we generated a heatmap of metabolites normalized to the control parental cell line (Figure 1D). We performed statistical analyses of these metabolites and found lipid metabolites to be significantly increased (Appendix A). In these analyses, lipids were again found to be enriched in BRAF V600E cells, and specifically, we found long chain PUFAs containing 18–22 carbons were highly abundant in cells expressing BRAF V600E. 

From these data, we identified that cells expressing BRAF WT have a different metabolic profile from cells expressing BRAF V600E. Further, lipid metabolites, specifically long chain PUFAs, were uniquely abundant in cells expressing BRAF V600E.

### 3.2. Cells Expressing BRAF V600E Do Not Exhibit Warburg-Like Metabolism

Next, we questioned how cells expressing BRAF V600E were using PUFAs. To address this question, we assessed metabolic flux using the live-cell metabolic Agilent Seahorse assay via a mitochondrial stress test. This mitochondrial stress test measures oxygen consumption rate (OCR) and extracellular acidification rate (ECAR) using the following metabolic poisons to disrupt the electron transport chain: oligomycin (complex V inhibitor), FCCP (proton uncoupler), antimycin (complex III inhibitor), and rotenone (complex I inhibitor) (Figure 2A). We performed a titration assay in 25 mM glucose to determine the optimal concentration of poisons to use for assessing mitochondrial function (Appendix A) and determined that 10 µM of each poison was most effective to measure mitochondrial function in our non-transformed NIH3T3 models. We next measured OCR and ECAR in BRAF WT, BRAF V600E, control, and parental cells when cultured in media supplemented with either glucose, palmitic acid, oleic acid, or α-linolenic acid. BRAF WT cells consistently displayed the highest OCR (Figure 2B–E) when cultured with glucose or lipids. In contrast, BRAF V600E cells had an OCR that was considerably reduced compared to BRAF WT cells and more similar to parental cells. Noticeably, ATP-linked production and maximal respiratory capacity spiked in BRAF WT cells when cultured in 25 µM oleic acid, but this was not observed for BRAF V600E or parental cells. Interestingly, BRAF WT cells had a high ECAR in media supplemented with 25 mM glucose and a low ECAR when cultured in palmitic acid, oleic acid, or α-linolenic acid (Figure 2F–I). BRAF V600E cells had a low ECAR in all cultured conditions, similar to control and parental cells. Thus, all cell lines appeared glycolytically inactive when cultured in the presence of palmitic acid, oleic acid, and α-linolenic acid. We found this result to be unexpected since both overexpression of BRAF WT and BRAF mutation are known oncogenic drivers and would be predicted to behave with a Warburg-like metabolism by relying on glycolysis to drive mitochondrial respiration [28,29].

In summary, these findings show that BRAF WT cells are more reliant on aerobic respiration than BRAF V600E or parental cells. In addition, BRAF WT cells are glycolytically active in glucose-abundant conditions. However, overexpression of either BRAF WT or BRAF V600E resulted in non-Warburg-like behavior with little glycolytic activity in lipid-abundant conditions. Taken together, these findings show BRAF WT cells are more sensitive to nutrient availability and have a flexible metabolism, whereas BRAF V600E cells are not only less flexible in their metabolism but are also less metabolically active than cells expressing BRAF WT.

### 3.3. BRAF V600E Expression Promotes Formation of Tunneling Nanotube (TNT)-like Protrusions Which Preferentially Accumulate Lipids

Our findings show that BRAF V600E cells are evidently consuming long chain PUFAs (specifically oleic acid and α-linolenic acid) for respiration at a slower rate than BRAF WT cells. Thus, we questioned where these lipids were located in the cell, and by using immunofluorescence with Nile red (lipid droplet staining dye) and phalloidin (that binds F-actin), we assessed lipid droplet localization (Figure 3A). These analyses revealed that BRAF V600E cells harbored significantly more lipid droplets than BRAF WT, control, and parental cells lines (Figure 3B). Notably, BRAF V600E cells had a unique morphology; specifically, BRAF V600E cells harbored lipid droplets that preferentially accumulated in long F-actin-containing protrusions (Figure 3C, D). We quantified the lipid droplets in this region of interest and compared it to the amount of lipid droplets in the perinuclear region (Figure 3E, F). These results demonstrated that lipid droplets preferentially accumulated in long F-actin-containing protrusions in BRAF V600E cells, but not BRAF WT or parental cells. Notably, similar F-actin-containing protrusions make up tunneling nanotubes (TNTs), which have been shown to have important roles in long-range cell communication by transferring cytoplasmic components between cells [30,31].

In summary, we found that overexpression of BRAF V600E changes the cell phenotype and accumulates lipid droplets in TNT-like protrusions.

### 3.4. Expression of BRAF V600E Enriches for Immunomodulatory Profiles

Thus far, our results have shown that long chain PUFAs were highly abundant in BRAF V600E cells that are not apparently used for metabolic respiration (Figure 2). Long chain PUFAs are metabolized from 18-carbon linoleic acid and α-linolenic acid to the longer 20-carbon PUFAs, arachidonic acid and eicosapentaenoic acid (Figure 4A) [32]. These 20-carbon PUFAs are able to serve as precursors for lipids mediators that play important roles in pro-resolving inflammation (also described as type II inflammation) [33]. Tandem mass spectrometry of the long chain PUFAs involved in synthesizing pro-resolving lipid mediators showed linoleic acid (LA), α-linolenic acid (αLA), arachidonic acid (AA), and eicosapentaenoic acid (EPA) were all highly abundant within cells expressing BRAF V600E, but not BRAF WT, control, and parental cells (Figure 4B). Notably, these long chain PUFAs were equally abundant in supernatants from cells expressing BRAF WT and BRAF V600E, but not control and parental cells (Figure 4C). Next, we measured the relative intracellular abundance of pro-resolving inflammatory fatty acids. While we were not able to measure all downstream lipid products, we determined the relative abundance of 9-hydroxyeicosatetraenoic acid (9-HETE), docosahexaenoic acid (DHA), and prostaglandin E2 (PGE2). Notably, Figure 4D–F reveal that these pro-resolving immunomodulatory lipids were highly abundant in BRAF V600E cells. To assess associated transcriptional changes, we performed quantitative real time PCR for genes associated with lipid and immune regulation (Figure 4G and Appendix A). *FAS* (Fas cell surface death receptor), *SCL27a1/FATP1* (solute carrier family 27 member 1), *CPT1a* (carnitine palmitoyltransferase 1a), and *PPARγ* (peroxisome proliferator-activated receptor γ) were exclusively upregulated in cells expressing BRAF V600E (Figure 4H–K).

These findings show pro-resolving (type II) inflammatory precursor PUFAs are highly abundant only in cells expressing BRAF V600E. Additionally, *BRAF V600E* mutation results in transcriptional upregulation of immunomodulatory and lipid regulators, which may play a role in tumorigenesis and inflammatory responses.

### 3.5. Circulating Plasma Lipids Are Increased in Melanoma Patients That Do Not Respond to MAPK Inhibitor Therapy

We next questioned if and how BRAFi therapy modulates the lipid profile in humans with advanced stage melanoma. To investigate this question, we collected plasma samples pre- and post-treatment with MAPK inhibitor (MAPKi) therapy from patients with late-stage melanomas that harbored a *BRAF V600E* mutation (Table 1). The patient cohort included responders (R) and non-responders (NR) to BRAFi/MAPKi (R is classified as a complete response or partial response; NR is classified as stable disease or progressive disease). We performed lipidomics on the plasma samples and completed unpaired analysis of responder vs. non-responder groups and paired analysis of pre- vs. post-treatment samples (Figure 5 and Appendix A, respectively). Long chain fatty acids, including palmitic acid, adrenic acid, and sphingomyelin, were elevated in the post-treatment plasma samples of non-responders. We also found the immunomodulatory lipid, lysophosphatidic acid, was also elevated in the post-treatment plasma samples of non-responders. Sphingomyelin was the only lipid found to be significantly lower in the plasma of non-responders from pre-treatment samples. Notably, sphingomyelin is the precursor to sphingosine-1-phosphate, which is a functional lipid mediator required for T cell homing and egress from secondary lymphoid organs [34]. Sphingosine-1-phosphate was not significantly different in either the unpaired analysis of responders and non-responders or the paired analysis comparing lipid levels pre- and post-treatment with BRAFi/MAPKi (Appendix A). However, myristoleic acid was found to be significantly elevated in post-treatment plasma samples of non-responders, whereas taurolithocholic acid was elevated in post-treatment plasma samples of responders. Altogether, these findings show that BRAFi/MAPKi treatment modulates systemic lipid levels in advanced stage cancer patients. Further, palmitic acid, adrenic acid, lysophosphatidic acid, and sphingomyelin could serve as potential markers for a response to BRAFi/MAPKi.

## 4. Discussion

This study examines how BRAF V600 status affects metabolism. Our findings identify key differences in cells overexpressing BRAF WT or BRAF V600E. Using an NIH3T3 overexpression model and tandem mass spectrometry, we show long chain PUFAs are enriched in cells expressing BRAF V600E, and that the immunomodulatory PUFA profile is determined by BRAF V600 status. We demonstrate metabolic consumption is different between cells expressing BRAF WT or BRAF V600E using the Agilent Seahorse assay, and cells expressing either BRAF WT or BRAF V600E exhibit non-Warburg-like behavior. We also report cells expressing BRAF V600E have a more elongated morphology and form TNT-like structures that specifically associate with lipid droplets. To our knowledge, this is the first report identifying immunomodulatory PUFA metabolism and TNT-like protrusion formation as unique characteristics of cells expressing BRAF V600E. We also demonstrate increases in immunomodulatory PUFA and long chain PUFA levels in the plasma of patients failing to respond to BRAFi therapy. In recent years, lipid metabolism and its role in cancer has been an emerging field that somewhat calls into question the Warburg effect [18].

The Warburg effect first established that cancer cells preferentially utilize glucose in a process known as aerobic glycolysis, even in the presence of oxygen [35,36,37,38]. Both overexpression of BRAF WT and BRAF V600E are reported as oncogenic events and would be predicted to promote a Warburg-like metabolism. Contrary to our hypothesis, we found both overexpression of BRAF WT and BRAF V600E in NIH3T3 cells resulted in little overall glycolytic activity when cultured in palmitic acid and other fats. While these cells overexpress cancer-promoting constructs, they may not be completely transformed cancer cells and may require a second “hit” to achieve a Warburg-like change in metabolism [39]. Additionally, the *BRAF V600E* mutation and its transformative potential is also controversial. The first report of *BRAF V600E* identified this alteration as a transforming mutation in NIH3T3 cells [6]. Yet, some non-transformed, healthy cell types, including normal nevi, express BRAF V600E and are benign [40,41,42]. Nevertheless, some cancers, such as hairy cell leukemia, almost exclusively express BRAF V600E mutations with few to no other known oncogenic drivers [12,43]. Our findings contribute to an evolving body of evidence aiming to elucidate the role of the Warburg effect and the transformative properties of BRAF expression.

The role of lipid trafficking in TNT-like structures in cancer remains unclear. We found that introducing the *BRAF V600E* mutation resulted in a phenotypic change with the formation of TNT-like structures that house lipid droplets. A recent study found myeloid cells use TNTs as a form of cellular communication [44]. Based on our findings and previous reports, we hypothesize cells expressing BRAF V600E have the potential to form TNT-like connections with surrounding cells to transfer communicating lipid droplets and pro-resolving fatty acids that may rewire the immune response for a tumor promoting response. Importantly, specific pro-resolving lipids, such as PGE2, can potentiate the suppressive function of myeloid-derived suppressor cells (MDSCs) [45]. Altogether, we suspect *BRAF V600E* mutation in certain cell types may rewire immune responses that are metabolically mediated by lipid signaling. We hypothesize lipid signaling may serve as a non-canonical second “hit” and ameliorate the inflammatory microenvironment. However, more investigation is needed to better understand how BRAF and lipid signaling may contribute to oncogenic and transformative potential in healthy cells and cancer cells.

Understanding the role of lipid signaling in cancer progression for BRAF-driven cancer cells is important for diagnosis, treatment, and improving patient outcomes. We found significant differences in the lipid profile and phenotype between cells expressing BRAF WT and BRAF V600E. Furthermore, we noted significant differences in plasma lipid levels between patients that responded and did not respond to BRAF inhibitor therapy. Lysophosphatidic acid, sphingomyelin, adrenic acid, and palmitic acid were all significantly increased in the plasma of BRAFi non-responders post-treatment. Notably, lysophosphatidic acid is a signaling lipid that is elevated in chronic inflammatory states, including cancer [46]. One prominent role of lysophosphatidic acid is rearrangement of the actin cytoskeleton [47]. In fibroblasts, lysophosphatidic acid induces actin polymerization resulting in the formation of cytoplasmic stress fibers that consist of filamentous actin (F-actin) and are associated with cell contraction, supporting movement, and migration of cells [47]. Furthermore, lysophosphatidic acid is involved in lymphocyte biology and has been shown to impair CD8 T cell anti-tumor immunity [46]. Specifically, lysophosphatidic acid signaling through its receptor (LPAR5) on CD8 T cells impairs intracellular calcium mobilization, disrupts T cell receptor stimulated ERK activation, and perforin degranulation [46]. Consequentially, CD8 T cell cytotoxic activity is significantly compromised in the presence of elevated lysophosphatidic acid levels. Further studies should elucidate the mechanisms whereby BRAF V600E cells that are resistant to BRAFi therapy increase lysophosphatidic acid production and whether targeting lipid signaling decreases treatment resistance.

While sphingomyelin increased post-treatment in patients that did not respond to BRAFi therapy, sphingosine-1-phosphate levels were not significantly different between BRAFi responders and non-responders pre- and post-treatment. Sphingomyelin is a precursor to sphingosine-1-phosphate, which promotes tumor growth via a number of mechanisms, including stimulation of G-protein coupled receptors and crosstalk with receptor tyrosine kinases [48]. Data in this study may be limited due to sample size, so further studies should investigate whether sphingosine-1-phosphate levels are altered based on BRAF status or BRAFi treatment resistance. Increased levels of adrenic acid and palmitic acid in non-responders post-BRAFi/MAPKi treatment may represent metabolic alterations in resistant tumors, such as increased fatty acid oxidation in response to metabolic stress induced by MAPKi. Previous studies have shown that melanoma cells increase CPT1a-dependent fatty acid oxidation in response to treatment with MAPKi therapy, and inhibiting MAPK, glycolysis, and fatty acid oxidation together inhibits tumor cell growth in vitro and in vivo [23]. Further, BRAF overexpression and mutation is observed across cancers and targeting metabolic pathways, including CPT1a, may be an alternative approach to improving the response to treatment [18,49]. In addition to MAPKi therapy, immunotherapy is another treatment option for melanoma. The data we have presented here shows that BRAF V600E expression modulates an immunomodulatory lipid profile. As such, levels of these lipids could impact the response to immunotherapy. More investigation in this area should be conducted to understand how signaling lipids modulate immune responses and therapeutic outcomes.

Further, it remains unknown how specific these metabolic features are to *BRAF V600E* mutation compared to other MAPK activating mutations. While there are similar lipid droplet phenotypes seen in diseases harboring *BRAF V600E* mutations [17], mutations in *RAS* are also frequently observed in melanoma and could potentially result in similar metabolic changes. This question about metabolic specificity is an open area for investigation.

Altogether, our study adds to the existing body of evidence that lipid signaling and fatty acid metabolism may be a potential therapeutic target for improving anti-cancer therapies. Our findings provide valuable insight into the clinical management and underlying pathophysiology of BRAF-driven tumors across cancers.

## 5. Conclusions

*BRAF* is frequently mutated across multiple cancer types and in normal nevi. Understanding how BRAF expression and mutation affects cellular metabolism, cytoskeletal structure, and oncogenic transformation is important for improving patient outcomes. The data presented here identifies BRAF as a key regulator of metabolism and cellular morphology. Further, we show lipid use and accumulation is in part determined by BRAF V600 status. Characterization of the *BRAF V600E* mutation in non-transformed cells showed that BRAF status modulates metabolite abundancy. Specifically, BRAF regulates the abundancy of immunomodulatory lipids and rewires metabolic flux for non-Warburg-like behavior. Cells exclusively expressing BRAF V600E remodeled the F-actin cytoskeleton to form long TNT-like protrusions, which specifically co-localized with lipid droplets. We further found that plasma lipid profiles in advanced stage melanoma patients were modulated by BRAF/MAPKi therapy. Immunomodulatory lipids, including lysophosphatidic acid, were elevated in patients that did not respond to targeted therapy. Altogether our findings show that BRAF V600 status plays an important role in regulating the immunomodulatory lipid profile that may offer potential therapeutic benefit and improve patient outcomes for cancers driven by BRAF.

## Figures and Tables

**Figure 1 cancers-14-02110-f001:**
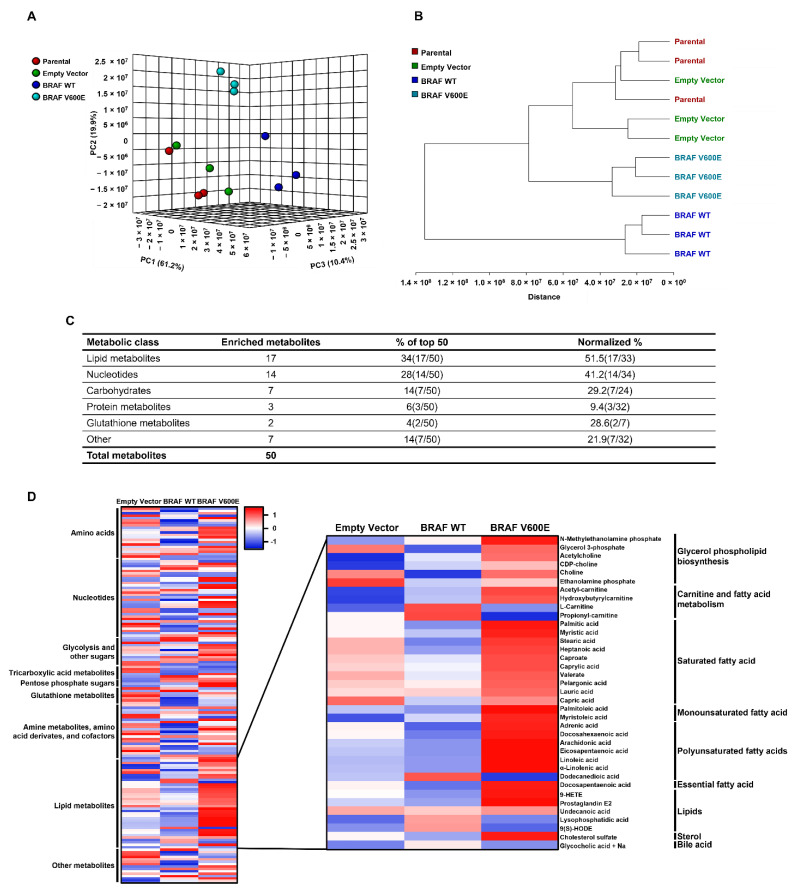
BRAF V600 status and expression modulates the metabolic profile. Global metabolomic data was collected by mass spectrometry for each biological replicate and evaluated by (**A**) unsupervised principal component analysis (PCA) and (**B**) and hierarchical clustering analysis. (**C**) Class and frequency of the top 50 metabolites present in BRAF V600E compared to BRAF WT. Normalized % refers to normalized abundancy based on the total number of metabolites in each class. (**D**) Heatmap representing relative metabolite abundancies with averaged triplicates normalized to the parental control. Z-scores represent highly abundant metabolites in red and less abundant metabolites in blue. Heatmap was generated using http://heatmapper.ca/ and was accessed on 19 February 2019.

**Figure 2 cancers-14-02110-f002:**
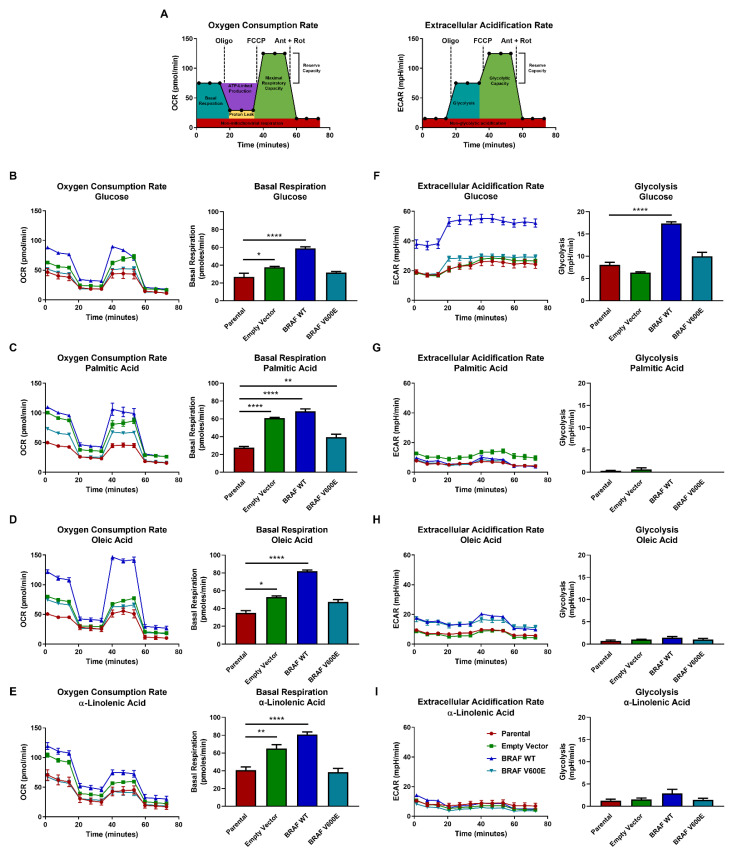
Cells expressing BRAF WT are more sensitive to nutrient availability than cells expressing BRAF V600E. (**A**) Schematics of Seahorse metabolic flux analysis for both oxygen consumption rate (OCR) and extracellular acidification rate (ECAR). The poisons used (10 µM each) are oligomycin (oligo), (4-(trifluoromethoxy) phenyl) carbonohydrazonoyl dicyanide (FCCP), antimycin (ant), and rotenone (rot). In OCR (left), red represents non-mitochondrial respiration, teal represents basal respiration, purple represents ATP-linked production, yellow represents proton leak, green represents maximal respiratory capacity, and the reserve respiratory capacity is calculated from [maximal respiratory capacity] − [basal respiration]. In ECAR (right), red represents non-glycolytic acidification, teal represents glycolysis, green represents glycolytic capacity, and the reserve glycolytic capacity is calculated from [glycolytic capacity] − [glycolysis]. OCR and ECAR were measured one hour after BRAF WT (blue lines), BRAF V600E (teal lines), control (green lines), and parental (red lines) cells were cultured in media supplemented with either (**B**,**F**) 25 mM glucose, (**C**,**G**) 25 µM palmitic acid, (**D**,**H**) 25 µM oleic acid, or (**E**,**I**) 25 µM α-linolenic acid. The ANOVA statistical test with post-hoc analysis was performed where * *p* < 0.05, ** *p* < 0.005, **** *p* < 0.00005 indicates the level of statistical significance.

**Figure 3 cancers-14-02110-f003:**
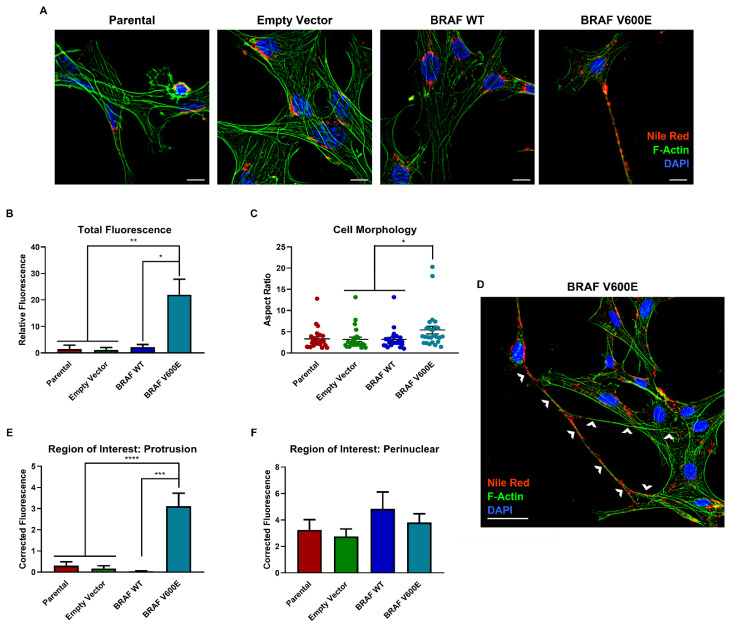
BRAF V600E overexpressing cells accumulate lipids in tunneling nanotube (TNT)-like structures. (**A**) Immunofluorescence staining for lipid droplets using Nile red (red), F-actin using phalloidin (green), and nuclei using DAPI (blue). Scale bars represent 20 µm. (**B**) Quantified total fluorescence in the red channel normalized to cell number. (**C**) Aspect ratios were determined by measuring the length and width of individual cells. (**D**) Annotated immunofluorescence staining from the experiment shown in panel (**A**) with white arrows highlighting TNT-like protrusions. (**E**,**F**) Regions of interest were quantified using fixed areas to measure fluorescence in the red channel. All quantifications were performed using FIJI and statistics were performed with GraphPad software. The ANOVA statistical test with post-hoc analysis was performed where * *p* < 0.05, ** *p* < 0.005, *** *p* < 0.0005, **** *p* < 0.00005 indicates the level of statistical significance.

**Figure 4 cancers-14-02110-f004:**
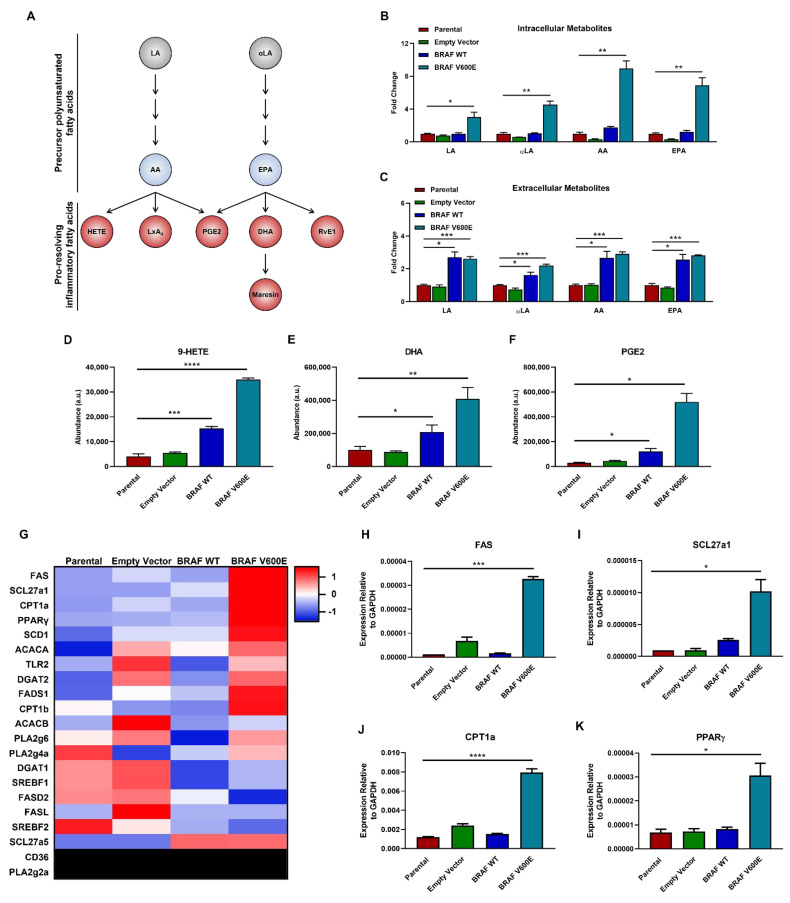
Immunomodulatory lipid enrichment and a unique transcriptional signature are exclusive characteristics of cells expressing BRAF V600E. (**A**) Schematic of polyunsaturated fatty acid metabolism. LA, linoleic acid; αLA, α-linolenic acid; AA, arachidonic acid; EPA, eicosapentaenoic acid; HETE, hydroxyeicosatetraenoic acids; LxA_4_, lipoxin A4; PGE2, prostaglandin E2; DHA, docosahexaenoic acid; RvE1, resolvin E1. Fold-change in abundancy of (**B**) intracellular and (**C**) extracellular LA, αLA, AA, and EPA in BRAF V600E, BRAF WT, and control cells normalized to parental cells. (**D**–**F**) Relative intracellular abundancies of (**D**) 9-hydroxyeicosatetraenoic acid (9-HETE), (**E**) docosahexaenoic acid (DHA), and (**F**) prostaglandin E2 (PGE2) in BRAF V600E, BRAF WT, and control cells normalized to parental cells. (**G**) Heatmap of 2 ^ (average of -dC_T_) values. Z-scores represent highly expressed mRNA in red and less expressed mRNA in blue. Heatmap was generated using http://heatmapper.ca/ accessed on 19 February 2019. Quantified mRNA expression for (**H**) *FAS*, (**I**) *SCL27a1*, (**J**) *CPT1a*, and (**K**) *PPARγ* by qRT-PCR analysis in BRAF V600E, BRAF WT, and control cells normalized to parental cells. Expression was normalized to *GAPDH* and the standard error of the mean of triplicates are represented by the error bars. Individual graphs represent the top four upregulated genes in cells expressing BRAF V600E. The ANOVA statistical test with post-hoc analysis was performed where * *p* < 0.05, ** *p* < 0.005, *** *p* < 0.0005, **** *p* < 0.00005 indicates the level of statistical significance.

**Figure 5 cancers-14-02110-f005:**
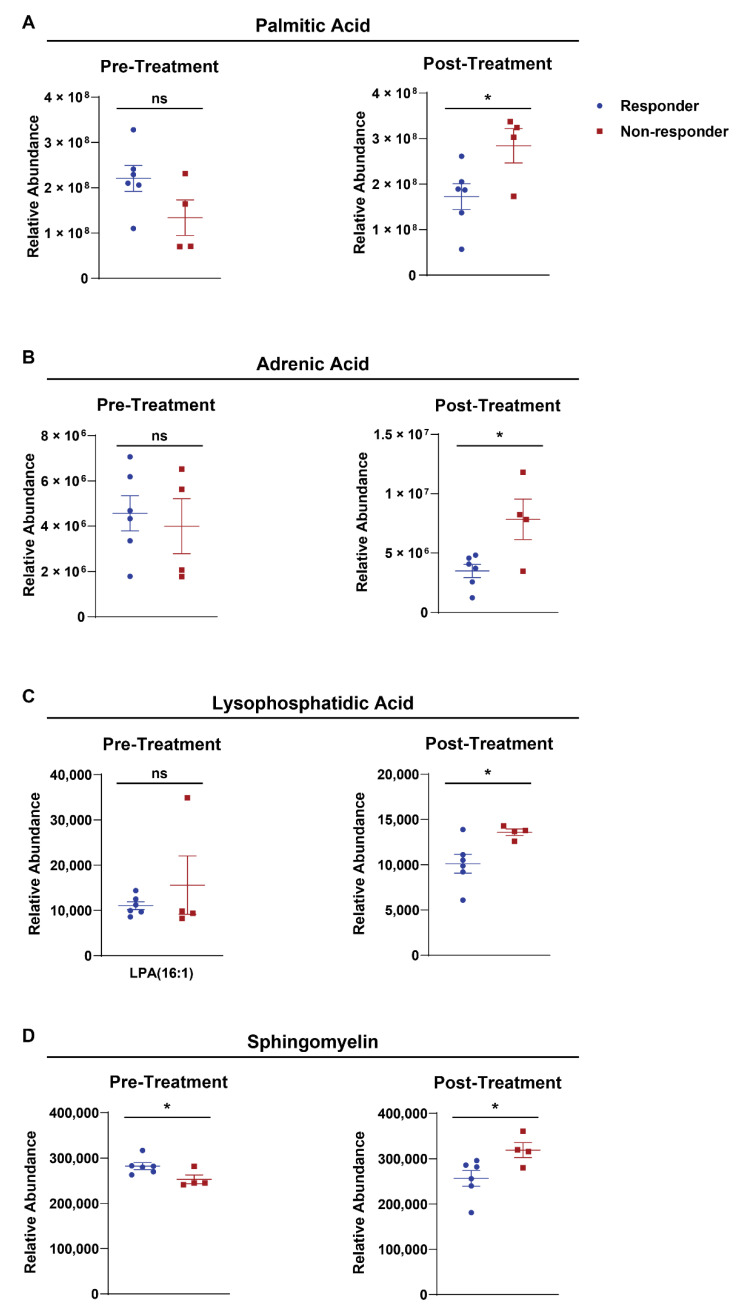
Plasma long chain fatty acid levels vary in response BRAF/MAPK inhibitor therapy in advanced stage melanoma patients. Relative abundance of (**A**) palmitic acid, (**B**) adrenic acid, (**C**) lysophosphatidic acid, and (**D**) sphingomyelin in responder patients (blue symbols; complete response and partial response) or non-responders (red symbols; stable disease and progressive disease) measured both pre-(left) and post-treatment (right). The unpaired Student’s *t* test analysis was performed where * *p* < 0.05 and ns designates not statistically significant.

**Table 1 cancers-14-02110-t001:** Patient clinical characteristics.

Patient No.	Sex	BRAF Status	Stage	Treatment	Response
1	F	V600E	IV	Dabrafenib + Trametinib	R
2	M	V600E	IV	Dabrafenib + Trametinib	R
3	M	V600E	IV	Dabrafenib + Trametinib	R
4	F	V600E	IV	Dabrafenib + Trametinib	R
5	M	V600E	IV	Dabrafenib + Trametinib	R
6	M	V600E	IV	Dabrafenib + Trametinib	R
7	F	V600E	IV	Dabrafenib + Trametinib	NR
8	F	V600E	III	Vemurafenib	NR
9	F	V600E	IV	Vemurafenib	NR
10	M	V600E	IV	Vemurafenib/Dabrafenib + Trametinib	NR/NR

## Data Availability

All data is made publicly available in the Appendix A.

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
