# Peer review of "BRAF Modulates Lipid Use and Accumulation"

_cancers, 2022, doi:10.3390/cancers14092110_

Round 1

Reviewer 1 Report

The authors have presented a comprehensive study investigating the role of BRAF modulating intracellular lipid accumulation and immune response. The study topic is very interesting and nicely designed experiment to support the conclusions.

It is suggested to break the results sections at each step. It will be easy for the authors to understand the article.

In the discussion the future perspectives sections seems very weak. The authors should have given discussion on the response to immunotherapy in given condition.

Author Response

Reviewer #1

The authors have presented a comprehensive study investigating the role of BRAF modulating intracellular lipid accumulation and immune response. The study topic is very interesting and nicely designed experiment to support the conclusions.

We wish to thank Reviewer #1 for their enthusiasm about on manuscript.

It is suggested to break the results sections at each step. It will be easy for the authors to understand the article.

We agree with the Reviewer, and we have modified the results section to be broken up at each step with subheadings to separate the findings from each figure.

In the discussion the future perspectives sections seem very weak. The authors should have given discussion on the response to immunotherapy in given condition.

As recommended, we have expanded our future perspectives in the discussion section. We have included a comment on how lipids profiles may affect therapeutic outcomes and response to immunotherapy in melanoma. The following was added on page 19, paragraph 2: “In addition to MAPKi therapy, immunotherapy is another treatment option for melanoma. The data we have presented here shows that BRAF V600E expression modulates an immunomodulatory lipid profile. As such, levels of these lipids could impact response to immunotherapy. More investigation in this area should be conducted to understand how signaling lipids modulate immune responses and therapeutic outcomes.”

Reviewer 2 Report

This is very interesting manuscript. The presented data shows that BRAFV600E oncogene can remodel lipids metabolism, what may have a crucial consequence in patients’ response to BRAF targeted therapies. It looks that this oncogene can oxidize fatty acids for energy. Authors found that BRAFV600E overexpressed in NIH/3T3 cells makes distinct lipids metabolism from NIH/3T3 cells overexpressed BRAFWT. This observation seems to be significant in context of what BRAFV600E does to normal cells to become tumor cells. They also postulating that targeting lipid metabolism may be another, complementary molecular target in anti-BRAF therapies.

However, the question remains, whether NIH/3T3 cells overexpressing, e.g. active Ras oncogene, would induce similar metabolic changes. In other words, how specific is the observed phenomenon for BRAFV600E and whether it depends also on a type of tumor cell with the BRAFV600E mutation. Maybe the authors have some data on this.

I would like to find al least some comments on such points in the manuscript discussion.

Author Response

We thank the Reviewers for their comments and helping us improve the quality of our manuscript. We have addressed each concern and incorporated changes to the manuscript, as described point-by-point below. Please note that references in this letter correspond to the clean version of the manuscript.

Reviewer #2

This is very interesting manuscript. The presented data shows that BRAFV600E oncogene can remodel lipids metabolism, what may have a crucial consequence in patients’ response to BRAF targeted therapies. It looks that this oncogene can oxidize fatty acids for energy. Authors found that BRAFV600E overexpressed in NIH/3T3 cells makes distinct lipids metabolism from NIH/3T3 cells overexpressed BRAFWT. This observation seems to be significant in context of what BRAFV600E does to normal cells to become tumor cells. They are also postulating that targeting lipid metabolism may be another, complementary molecular target in anti-BRAF therapies.

We thank the Reviewer for their detailed analysis of our manuscript.

However, the question remains, whether NIH/3T3 cells overexpressing, e.g. active Ras oncogene, would induce similar metabolic changes. In other words, how specific is the observed phenomenon for BRAFV600E and whether it depends also on a type of tumor cell with the BRAFV600E mutation. Maybe the authors have some data on this.

I would like to find at least some comments on such points in the manuscript discussion.

We agree with the Reviewer that this question of specificity is quite interesting.

We have updated our discussion to include comments about this on page 19, paragraph 3: “Further, it remains unknown how specific these metabolic features are to BRAF V600E mutation compared to other MAPK activating mutations. While there are similar lipid droplet phenotypes seen in diseases harboring BRAF V600E mutations [17], mutations in RAS are also frequently observed in melanoma and could potentially result in similar metabolic changes. This question about metabolic specificity is an open area for investigation.”